# The Influence of A Mo Addition on the Interfacial Morphologies and Corrosion Resistances of Novel Fe-Cr-B Alloys Immersed in Molten Aluminum

**DOI:** 10.3390/ma12020256

**Published:** 2019-01-14

**Authors:** Zicheng Ling, Weiping Chen, Weiye Xu, Xianman Zhang, Tiwen Lu, Jian Liu

**Affiliations:** 1Guangdong Key Laboratory for Advanced Metallic Materials Processing, South China University of Technology, Guangzhou 510640, China; lingzichengbyd@163.com (Z.L.); xu782673255@126.com (W.X.); lutiwen_scut_15@163.com (T.L.); rjliu1210@163.com (J.L.); 2Mechanical and Electrical Engineering College, Hainan University, Haikou 570228, China; xianman213@163.com

**Keywords:** Fe-Cr-B alloys, borides, weight loss, interface, EPMA

## Abstract

The influence of a Mo addition on the interfacial morphologies and corrosion resistances of novel Fe-Cr-B alloys in molten aluminum at 750 °C was systematically investigated using scanning electron microscopy, X-ray diffractometer, electron probe microanalysis, and transmission electron microscopy. The results indicated that Mo could not only strengthen the matrix but also facilitate the formation of borides. Furthermore, the microstructures of Mo-rich M_2_B boride changed from a local eutectic net-like structure to a typical coarse dendritic structure and a blocky hypereutectic structure with increasing Mo addition. This was true of the blocky Mo-rich M_2_B boride, rod-like Cr-rich M_2_B boride and the corrosion products, which had a synergistic effect on retarding of the diffusion of molten aluminum. Notably, the corrosion resistance of the Fe-Cr-B-Mo alloy, with an 8.3 wt.% Mo addition, was 3.8 times higher than that of H13 steel.

## 1. Introduction

It is acknowledged that molten aluminum is one of the most aggressive metal melts due to its high chemical activity it can inevitably react with most other metals [1,2]. In the aluminum industry, especially in cast dies and hot dip aluminizing production process, a wide range of metallic components, such as crucibles, die-casting molds and charging cylinders, are in contact with molten aluminum or its alloys. These components suffer from serious corrosion while melting and can even fail [3,4]. Thus, frequently changing these components wastes money and time. Therefore, a larger number of works [5,6,7,8] have been conducted to explore new materials with high corrosion resistance to molten aluminum. Meanwhile, the corrosion mechanism of steels in molten aluminum has also been widely studied in recent years [9,10,11]. The corrosion mechanism of steel in molten aluminum can be summarized by the following two points: one is the dissolution of iron and steel the other is the nucleation and growth of the Fe-Al intermetallic compounds (IMCs), such as the phase of η-Fe_2_Al_5_ and θ-FeAl_3_ [12]. Additionally, the corrosion process between steel substrate and molten aluminum is mainly controlled by the inter-diffusion of Fe and Al atoms [12] and the driving force is the different chemical potential between the steel substrate and the molten aluminum [1]. Thus, limiting the atomic inter-diffusion between the substrate and molten aluminum plays a key role in ameliorating the materials’ corrosion resistances. Additionally, the cracks in the corrosion layer, resulted from the phase transition stresses and the different thermal expansion coefficients among the IMCs also lead to corrosion product spalling [13]. 

Previous studies have proposed methods to increase the corrosion resistance and operating life of cast. There are certain die materials with surface treatment technology, such as surface coating [6,14], surface alloying [7,15], and surface duplex treatments [16,17]. However, surface treatments are restricted to the surface coating uniformity, the thermal expansion properties, and the wettability similarity with substrate materials. However, the brittle and thin protective coating layer is susceptible to rapid corrosion and even abrasion in molten aluminum, which may be unavoidable to spalling and may even lead to failure. Recently, much attention has been paid to Fe-B alloys due to their unique net-like structures and properties. For example, in related studies, cast B-added steel has a high hardness and corrosion resistance to molten zinc due to its 3D net-like boride [18,19]. Additionally, it has proved that boronising on steel and iron can improve the corrosion resistance in molten aluminum because of the formation of Fe_2_B or Fe_2_B/FeB [20]. However, the intrinsic brittleness of the Fe_2_B phase restricts its related material development and applications. 

To provide insight into the abovementioned problem, several attempts [21,22,23,24,25,26,27,28] have been made to modify the undesired microstructure by alloying with metals, such as Cr [21,22], Ti [23,24], W [25], and Mo [26], to stabilize the M_2_B-type borides. Among these metals, Mo not only has a low solubility in molten aluminum [27], but also increases the hardenability via co-addition into the B-added steels [28]. Although the Fe-Cr-B-Mo cast steels with different B additions were studied in our previous study [8], the role of how different Mo additions affect the interfacial morphologies and corrosion resistances of Fe-Cr-B alloys in molten aluminum is still unknown. Therefore, the present work aims to prepare a kind of novel Fe-Cr-B alloy with different contents of Mo and their interfacial morphologies and corrosion resistances in molten aluminum were also investigated.

## 2. Materials and Methods

### 2.1. Material Preparation

The investigated novel Fe-Cr-B alloys with different contents of Mo were melted in a 250 kg–capacity medium-frequency induction furnace. The initial charge materials were clean A3 steel and ferroalloys, such as Fe-70 wt.% Cr and Fe-60 wt.% Mo; then, Fe-32 wt.% B was added into the slag to minimize the oxidation loss. When all the alloys were melted, they were then transferred into the pre-heated teapot ladle with a small amount of pure Al to deoxidize at 1530 °C. After removal of the slag, the high-temperature melt was poured into sodium silicate-CO_2_ bonded sand molds at 1450 °C, obtaining the Y-block ingots following ASTM A781/A 781-M95. Notably, all the prepared alloys were adopted the multiple pouring in one furnace by adding the charge materials. After these ingots were cooled to room temperature, they were cut from the lower part of the Y-block ingots using a wire-electrode with dimensions of 10 mm × 10 mm × 10 mm. Additionally, the H13 steel was selected as a standard sample. The chemical compositions of the designed alloys were analyzed using inductively coupled plasma atomic emission spectroscopy (ICP-AES); the results for the H13 steel are also given in Table 1. 

### 2.2. Material Preparation

All surfaces of the test samples were ground using 1500 mesh SiC paper, washed with ethanol in an ultrasonic cleaner and then weighted on an electronic balance prior to the corrosion test. A schematic diagram of the corrosion tester is shown in Figure 1. First, five surfaces of the test samples were coated with alumina and then wrapped in graphite paper before being inserted into a graphite mold. Second, the uncovered surfaces were exposed to the 3.0 kg unsaturated molten aluminum (99.99 wt.%) at 750 °C for various immersion times (0.5 h, 1 h, 4 h, 6 h, and 8 h). Additionally, three parallel samples of each type alloy were used. During the corrosion test experiments, to monitor the process temperature, a K-type thermocouple covered by alumina was immersed into the molten aluminum. Finally, before determining the weight loss, the air-cooled samples were put into a 10% NaOH solution to remove the residual molten aluminum. In this study, the corrosion rates of the samples were assessed by the weight loss per square centimeter of the sample per minute, i.e., (mg·cm^−2^min^−1^).

### 2.3. Microstructure Characteristics

Cross-sections of the corroded samples were cut perpendicular to the corrosion layer, followed by a standard grinding and polishing for the metallographic examinations. Additionally, the polished samples were etched in a solution (4 mL HNO_3_ + 4 mL HF + 92 mL H_2_O). The microstructures of the as-cast specimens and corroded specimens were observed using scanning electron microscopy (SEM, Carl Zeiss NTS GmbH, Oberkochen Germany) equipped with a back-scattered electron (BSE). The phase analyses of the designed Fe-Cr-B alloys and the interfacial corrosion products were performed using an X-ray diffractometer (XRD, Bruker D8 ADVANCE, Karlsruhe, Germany) with Cu Kα radiation. Besides, the chemical compositions of the different phases of the Fe-Cr-B alloys and corrosion layers, electron probe microanalyses with a wavelength dispersive X-ray (WDX) analysis (EPMA, JXA-8230, JEOL, Tokyo, Japan) were performed. To further understand the phase compositions of pristine matrix and the corrosion products, thin foil samples were prepared by mechanical thinning followed by ion milling, then which were observed under a transmission electron microscope (TEM, TECNAL G2 S-TWIN F20, FEI, Hillsboro, USA) with selected area electron diffraction (SAED) analysis and an energy dispersive spectrometer (EDS).

## 3. Results and Discussion

### 3.1. As-Cast Microstructure

The XRD patterns and microstructures of the Fe-Cr-B alloys are shown in Figure 2 and Figure 3, respectively. According to the XRD results (Figure 2), it can be identified that the main phases of M1-M4 are α-(Fe, Cr), M_2_B-type borides. As shown in Figure 3a–c, the morphologies of these borides display rod-like, dendritic and net-like structures. As shown in Figure 3c, when the Mo content reaches to 6.19 wt.%, the interesting dendritic eutectic borides extensively appear. Moreover, when the Mo content reaches to 8.30 wt.%, a new kind of irregular blocky boride is detected in M4 (Figure 3d). It may be inferred that the blocky borides are the hypereutectic Mo-rich boride. It is clearly found that rod-like borides decrease as the addition of Mo increases (Figure 3b–d). In addition, net-like structures are adjacent to the rod-like borides in some light regions (Figure 3b). Additionally, from the XRD pattern of M4 sample, it can be observed that carbide peaks are more intense than that of M1-M3, which may be inferred that there are more carbides formation owing to the highest content among the designed alloys. Moreover, the chemical compositions of the various borides are measured using EPMA and are shown in Table 2. The results suggest that the rod-like Cr-rich boride (Figure 3a, point 1) has a stoichiometry of Fe1.12Cr0.88B, which is close to the Fe1.1Cr0.9B0.96 phase reported in the literature [29]. The net-like Cr-rich borides (Figure 3a, point 2 and Figure 3b, point 3) have a stoichiometry of Fe1.34Cr0.68B and Fe0.80Cr0.55Mo0.53B, respectively. In contrast, the stoichiometry of the dendritic boride (Figure 3c, point 4) and the blocky boride (Figure 3d, point 5) are Fe0.82Cr0.64Mo0.78B and Fe0.59Cr0.47Mo0.64B, respectively. 

Figure 4 shows the bright-field TEM micrographs and corresponding selected area diffraction patterns (SADPs) of borides and matrix in alloy M4. From the Figure 4a,d, it can be noticed that the primary rod-like Cr-rich M_2_B boride is a body-centered orthorhombic (BCO) structure (such as CrFeB as detected by XRD with the lattice parameters of a = 1.4534 nm, b = 0.7302 nm, c = 0.4215 nm), which is different from the single phase Fe_2_B with a body-centered tetragonal (BCT) structure with the lattice constant of a = b = 0.5109 nm, c = 0.4249 nm, c/a = 0.832. Notably, it has been reported that when the Cr content is higher than 8 wt.%, the phase of the boride transforms from tetragonal (Fe, Cr)_2_B to orthorhombic (Cr, Fe)_2_B [30]. The SADPs taken from the eutectic net-like M_2_B boride (Figure 4b,e) reveals a BCO structure, which is in agreement with the literature [29]. The TEM analysis on the matrix reveals a body-centered cubic (BCC) structure (Figure 4b,f). It can be inferred that it is a typical ferritic structure with supersaturated alloying elements such as Cr, Mo, and little of C and B. Additionally, the TEM analysis on SADPs in Figure 4c,g indicates that the blocky Mo-rich M_2_B boride can be indexed to a BCT structure with the lattice parameters of a = b = 0.5547 nm, c = 0.4739 nm, c/a = 0.85. The interplanar distance of the (110) plane for Mo_2_B is 0.4012 nm, which is in agreement with the XRD results (d(110) = 0.3900 nm). Obviously, the quantitative elemental analysis of various morphologies of borides are confirmed to be M_2_B-type borides. All the WDX analysis results on the various borides are in agreement with the previous TEM analysis with the corresponding SADPs. 

The formation of M_2_B-type borides is ascribed to the low solution of B in austenite/ferrite and the non-equilibrium segregation of B along the austenite grain boundary during the solidification process [31,32]. Actually, without Mo addition, owing to the similar crystal structural parameters of Fe and Cr atoms, Cr atoms may first substitute for Fe atoms and form the eutectic net-like and even the rod-like Cr-rich boride during the solidification, which is also conformed to the solidification rule of Fe-Cr-B-C phase diagram [30,33]. Then, with the Mo addition, the nucleation and growth of eutectic net-like Mo-rich borides may be followed by depleting the adjacent Cr and B atoms [29]. Applying more Mo addition (when the content reaches to a certain value, which is more than 6.3 wt.%) to the alloy, it occurs the hypereutectic blocky Mo-rich M_2_B-type boride. Obviously, superfluous Mo atoms would may also dissolve in carbide and ferrite.

### 3.2. Corrosion Resistance

Figure 5 shows the weight loss rates of the designed Fe-Cr-B alloys and H13 steel immersed in the molten aluminum. Obviously, it can be seen that all the Fe-Cr-B alloys exhibit comparatively better corrosion resistances than that of H13 steel. Except for the immersed times of 0.5 h and 1 h, the variation trends of the curves of M1-M4 samples decrease with the immersion time, which are different from that of H13. Additionally, the result shows that the weight loss rate of the M1-M4 samples with the Mo addition presents a decreasing trend, except for the M3 sample. However, the M1 sample presents a good corrosion resistance, which may be ascribed to the comparatively higher B content (which can form more M_2_B-type boride) than that of M2-M3. Among them, the corrosion resistance of M4 sample is 3.8 times higher than that of H13 steel at 8 h. Therefore, it can be inferred that the various borides and their morphologies can play a key role in improving the corrosion resistance of Fe-Cr-B-Mo alloys in molten aluminum.

### 3.3. Cross-Sectional Morphology of the Interfacial Corrosion Layer

The cross-sectional interfacial morphologies of M1-M4 alloys and H13 steel corroded by molten aluminum are shown in Figure 6. The images show that the interfacial morphology of H13 steel (Figure 6e) immersed in molten aluminum is different from that of Fe-Cr-B alloys with various Mo contents (Figure 6a–d). Obviously, the interfacial IMC layers of the M1-M4 samples are composed of some special periodic layered structures (PLSs) (Figure 6a–d). According to the corrosion rate of M1-M4 (Figure 5), it can be inferred that the PLSs can hinder the atomic inter-diffusion between the substrate and the molten aluminum. However, as the corrosion progress goes on, the PLSs can be corroded into particles (close to molten aluminum). Notably, the interfaces between the M1-M4 substrates and the IMC layers are smoother than that of H13 steel. The interfacial morphology of H13 steel (Figure 6e) presents one kind of thick finger-like Fe-Al IMC growing up towards the steel matrix and the other kind of thin needle-shaped Fe-Al IMC adjacent to the molten aluminum. Notably, there are quantity of cracks in the corrosion layer (Figure 6e). These micro-cracks are very harmful to the matrix, which will allow the molten aluminum to more easily infiltrate into the substrate and will hence result in reactions causing the interfacial layer to spall off and break down [34]. For the alloys of M3 and M4 especially, the exfoliated Mo-rich M_2_B borides are also found (Figure 6b–d). Additionally, there are few micro-cracks in the corrosion layer, and the number of micro-cracks decreases with the increase in Mo addition (Figure 6a–d). As previously mentioned, the phase transition in the corrosion layer and the different thermal expansion coefficient among the various phases result in the formation of micro-cracks [35,36,37].

The high magnifications illustrated in Figure 6a–d show that the matrixes are corroded first. Meanwhile, the rod-like (Cr, Fe)_2_B and Mo-rich M_2_B borides extrude into the corrosion layer (Figure 6a–d), which proves that the borides can withstand the corrosion of molten aluminum for a much longer time than H13 steel and can present a better corrosion resistance [8,35]. Additionally, the morphology of boride plays a significant role in improving the corrosion resistance in molten aluminum, especially for the Mo-rich M_2_B-type boride, from the weight loss rates of the M2-M4 samples (Figure 5). Combining the related cast microstructures (Figure 3b–d), it can be inferred that the net-like Mo-rich borides (Figure 6b,c) cannot withstand molten aluminum for a long time because of their small size. However, the M4 sample presents the best corrosion resistance among all of the samples, which is ascribed to the blocky Mo-rich boride structure (Figure 6d) with its much larger size.

As observed in Figure 7, the relationship between the thicknesses of IMCs and the corrosion times of sample M4 and H13 steel indicates that the curves generally follow a nonlinear law and that the IMC thickness of M4 alloy is much larger than that of the H13 steel for all immersion times. The IMC thickness of M4 (~1437 μm) is 10.8 times than that of H13 steel (~133 μm) after being immersed for 8 h. However, the thickness of the corrosion layer experiences variations of increases and decreases due to the complex interfacial reaction and spallation mechanism. The thickness of IMC layer of M4 sample increases with the increase in immersion time. The reason may be ascribed to the following aspects: On one hand, the 3D net-like structure of borides can protect the Fe-Al IMCs from spalling. On the other hand, the IMC layers can serve as barriers to weaken the inter-diffusion rate of molten aluminum and matrix, which leads to the improvement of the corrosion resistance [34,38]. However, when the immersion time is longer than 4 h, the intermetallic thickness of H13 steel is decreased. The most important reason to explain the phenomenon is the initiation and propagation of micro-cracks in the corrosion layer, which accelerates the IMC corrosion and spallation. The results are consistent with the weight loss rates, as shown in Figure 5.

### 3.4. Corrosion Products

As shown in Figure 6a–d, PLSs obviously appear between Cr-rich borides and molten aluminum. To understand the element distributions in PLS, the EPMA line analysis of a typical sample of M3 along the line segment AE is presented in Figure 8. The results indicate that the Cr and Mo elements are mostly distributed in the borides (Figure 8a,c,e). The concentration variations of the Cr, Mo and B elements are different from those of Fe in the IMC layers. Additionally, the PLSs are generated at the interface between the molten aluminum and the Cr-rich borides (Figure 8a,c,d) rather than at the Mo-rich boride interface (Figure 8b–d). It can be inferred that the interfacial reaction takes place between the Cr-rich boride and the molten aluminum. In our previous study [34], the Cr-rich boride reacted with molten aluminum through the following reaction: (Cr, Fe)_2_B + Al → Cr-Al-B + Fe-Al. Although the Cr-rich boride and the Mo-rich boride present a superior corrosion resistance to molten aluminum, both were eventually corroded and were then embedded in the Fe-Al IMCs, playing a key role in protecting the Fe-Al IMC from spalling off.

To explore the corrosion products of the different borides of M4 alloy immersed in molten aluminum at 750°C for 8 h, the XRD and SEM-EDS analysis results are used to characterize these phenomena. From the XRD pattern of the corrosion layer (Figure 9a), it can be clearly found that the corrosion layer mainly consists of (Fe, Cr, Mo)_4_Al_13_, (Fe, Cr, Mo)_2_Al_5_ and exfoliated M_2_B-type boride. Figure 9b shows that net-like eutectic Cr-rich borides are present the PLSs morphology under the attack of molten aluminum. This kind of PLSs is tiny and local, which is different from those of the samples of M1-M3 (Figure 6a–c) that were corroded in molten aluminum. Additionally, the PLSs are changed into tiny particles and even dissolve under the long reaction time and thermal attack from the molten aluminum (Figure 9d,e). By observing Figure 9c, it can be seen that the blocky Mo-rich M_2_B boride phase distributes into the corrosion layer (owing to the corrosion of the supporting point of matrix) and keeps the original form, which indicates that this phase has outstanding corrosion resistance in molten aluminum. From the Mo-rich borides in the corrosion layer, it can be inferred that the Mo element can enhance the thermal stability of Fe_2_B in molten aluminum. It has been reported that Cr and Mo atoms can substitute Fe atoms and can form M_2_B-type borides, which could improve the fracture toughness of the Fe_2_B phase [21,22,27,39]. Additionally, from the longitudinal interfacial corrosion layer (Figure 9d), it also can be observed that there are some grainy corrosion products. Figure 9e is the high magnification view of these grainy zones marked with rectangle. The average composition of the grainy zone at point 1 measured by WDX is as follows: 39.34 wt.% Cr, 50.56 wt.% Al, 7.72 wt.% B and 2.35 wt.% Fe. It can be inferred that they are the Cr-Al-B IMCs. From the line scanning of these zones in Figure 9f, it can be easily found that the content of Cr and Fe shows an opposite trend of variation, which also proves the composition of the grainy phase.

To further understand the phase composition, the Bright-field TEM micrographs and corresponding SADPs of corrosion products were also be conducted. The corresponding chemical compositions of all the phases are shown in Table 3. The TEM analysis on SADPs taken from columnar zone (Figure 10a,c) can be indexed to a monoclinic structure, and it belongs to C2 space group and Fe_4_Al_13_-type structure (a = 1.5489 nm, b = 0.8083nm, c = 1.2476 nm). The interplanar distance of the (200) plane for (Fe, Cr, Mo)_4_Al_13_ is 0.7410 nm, which is also in good agreement with the XRD results (d(200) = 0.7378 nm). From the Figure 10b, it can be seen some corrosion products are roughly parallel to each other, and the corresponding SADPs result (Figure 10d) reflects a typical polycrystalline area. According to the chemical composition analysis of E region (Table 3), it is obvious that they are the Cr-Al-B intermetallic. It may be inferred that they belong to the PLSs corrosion products produced from the Cr-rich M_2_B boride. However, according to the isothermal section of Cr-Al-B system [40], it was acknowledged that there were three ternary stoichiometric intermetallics, AlCr_2_B_2_, AlCr_3_B_4_ and AlCr_3_B_7_. Therefore, the further study will be needed to clarify the formation mechanism of PLS and the crystal structure of Cr-Al-B intermetallic.

Figure 11 shows the elemental X-ray mapping distribution of M4 after the corrosion test for 8 h at 750 °C, as analyzed using EPMA analysis. The results clearly show the distribution of Fe, Cr, B, Mo, and Al atoms. Observing the (Cr, Fe)_2_B (Figure 11b–d) and Mo-rich M_2_B (Figure 11c–e) reveals that both of them have the effect of retarding the corrosion rate of molten aluminum by functioning as diffusion barriers. Similarly, Mo-rich and Cr-rich borides also encounter the dissolution, diffusion, and spallation stages as the immersion time extends. As observed in Figure 11a,c,d, the tiny PLSs contain atoms of Cr, Al, and B, which agrees with the EPMA (Figure 8e,f) and TEM analysis results (Figure 10b,d). As previously mentioned, observing Figure 11c–f evidently shows that the Mo-rich borides have an excellent corrosion resistance. On the one hand, these borides can protect the substrate from the high temperature molten aluminum diffusion and dissolution (Figure 11a,e,f). On the other hand, when the substrate is corroded, the Mo-rich borides will lose the attachment points and will drop into the molten aluminum, forming the transition layer. The combination mode of Mo-rich borides and molten aluminum is not a kind of physical bond but is a metallurgical bond, resulting in the formation of Al-B polar covalent bonds and Al-Mo metallic bonds [41]. This kind of bonding has a strong binding force, which likely hinders the IMC from falling off. Additionally, Figure 11a,f shows that there are no Al elements in the cracks, which indicates that the formation of cracks does not occur during the immersion process or that it may occur during the solidification and machining processes. In light of these observations, the Cr-rich borides and Mo-rich borides may play a synergistic role in discouraging the molten aluminum from further diffusion and reducing the interfacial corrosion rate of the metal substrate.

The Nernst-Shchukarev equation [42,43], which describes the dissolution of a solid metal in a molten metal, can be written as
dC/dt = K·A/V·(Cs − C)(1)
where C is the instantaneous concentration of the dissolved metal in the melt, Cs is the saturation concentration, K is the dissolution rate constant, A is the surface area of the solid metal, and V is the volume of the melt. In the integrated form, Equation (1) becomes (initial conditions: C = 0, t = 0)
C = Cs[1 − exp(−K·A·t/V)](2)
Equations (1) and (2) indicate that if the volume of the molten metal is small, the dissolution of the solid metal will increase, and then the dissolution rate dC/dt will decrease. In contrast, if the volume of molten metal is larger, as in an industrial hot-dip bath, the dissolution rate dC/dt will be constant [43]. As is reported in the literature [28], for the corrosion of Mo, Nb, Cr, and Y that is immersed in a small volume of molten aluminum, the dissolution rate dC/dt decreases with time, which is also consistent with the results in this research (Figure 6).

### 3.5. Corroded Surface

Figure 12a–e shows the morphologies of the corroded surfaces of M1-M4 alloy and H13 steel after removing the residual aluminum using a 10% NaOH solution and Figure 12f shows the morphology after the removal of the intermetallic layer of M4 alloy. It can be seen that the morphologies of M1-M4 alloy and H13 steel are obviously different from each other. Some columnar structures (where the arrows are pointed in Figure 12) appear in the IMC layers. It may be inferred that they are the corroded rod-like (Cr, Fe)_2_B. Additionally, the PLSs are also shown in Figure 12b, which are embedded into the Fe-Al IMCs. However, owing to the rough surface, not every corrosion layer that can be found in the PLSs. As observed in the Mo addition alloys, M2-M4, especially M4 (Figure 12d), more columnar structures appear in the IMC layer, and they can act as nails to protect the Fe-Al IMCs from spalling off. Interestingly, from the high magnification views that are marked by white rectangles, there exist some fold structures of Mo-rich M_2_B in the Fe-Al IMC layer. Similarly, to a certain extent, these fold structures can hinder the atomic inter-diffusion of the substrate and molten aluminum. In contrast, the corroded surface morphology of H13 has many micro-cracks (Figure 12e). These micro-cracks exist and can act as liquid channels to accelerate the substrate corrosion and the spallation of Fe-Al IMCs under the attack of molten aluminum [13]. Furthermore, after removing the corrosion layer of the sample M4, it can be clearly found that numbers of rod-like (Cr, Fe)_2_B and the blocky Mo-rich M_2_B, which are similar to “tree roots”, fasten the matrix tightly (Figure 12f), which can contribute to hindering the matrix corrosion effectively.

Based on the previous investigation of Fe-Cr-B alloys in molten aluminum, the corrosion process can be divided into the following three processes: the dissolution of matrix, formation, and spallation of Fe-Al IMCs at the interface, and the spallation of the boride phase. On the basis of the experiment results, the novel Fe-Cr-B-Mo alloy with 8.3 wt.% Mo addition exhibits a better corrosion resistance to molten aluminum than that of H13 steel. The reasons for this result may be ascribed to the following two points. First, the M_2_B-type boride, especially the rod-like (Cr, Fe)_2_B and the blocky Mo-rich M_2_B boride, show a slower reaction rate with molten aluminum. Second, both borides can act as roots to capture the Fe-Al IMCs and prevent them from spalling off.

## 4. Conclusions

This work mainly investigated the influence of Mo addition on interfacial morphologies and the corrosion resistances of Fe-Cr-B alloys in static molten aluminum at 750 °C. The results are summarized as follows:(1)Mo addition plays an important role in the morphology of M_2_B-type borides. The microstructures of Mo-rich M_2_B borides change from a local net-like eutectic structure to a typical coarse dendritic structure and appear a kind blocky hypereutectic structure with the increase of Mo content.(2)The corrosion resistance of the M4 sample with an 8.3 wt.% Mo content is 3.8 times higher than that of H13 immersed in molten aluminum due to its blocky Mo-rich M_2_B boride and rod-like (Cr, Fe)_2_B.(3)The various M_2_B-type borides phases result in a synergistic effect that retards the corrosion of molten aluminum and presents a much slower corrosion rate than the matrix.(4)The Mo-rich M_2_B boride, rod-like (Cr, Fe)_2_B and its corrosion products of PLSs can act as roots to capture the Fe-Al IMCs and to prevent them from spalling off.

## Figures and Tables

**Figure 1 materials-12-00256-f001:**
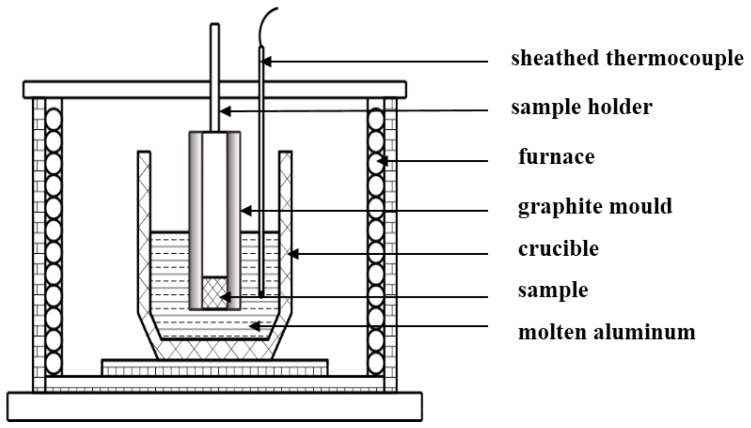
Schematic diagram of the corrosion test device.

**Figure 2 materials-12-00256-f002:**
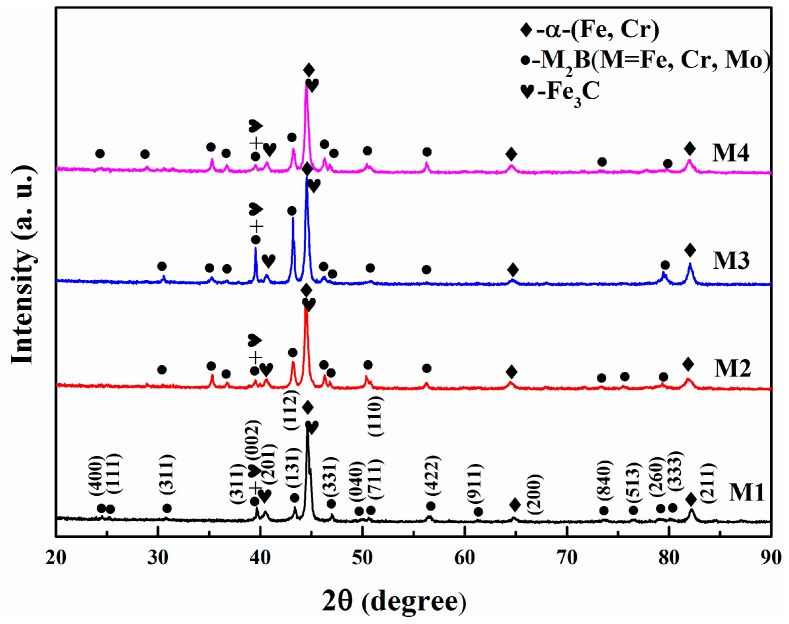
X-ray diffractometer (XRD) patterns of the as-cast Fe-Cr-B alloys with various Mo additions.

**Figure 3 materials-12-00256-f003:**
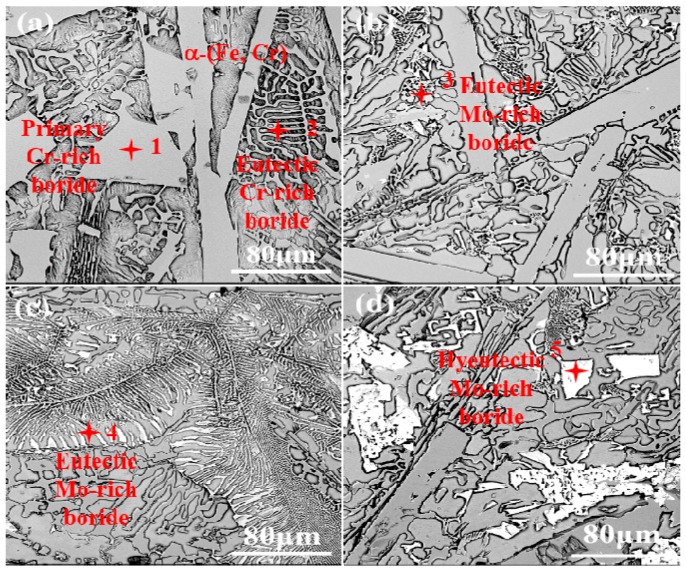
Back-scattered electron (BSE) images of the Fe-Cr-B alloys with various Mo additions: (**a**) 0 wt.% Mo; (**b**) 3.22 wt.% Mo; (**c**) 6.19 wt.% Mo; (**d**) 8.30 wt.% Mo.

**Figure 4 materials-12-00256-f004:**
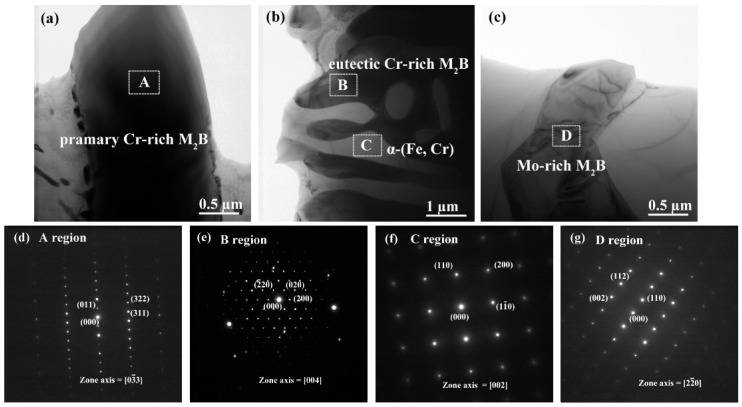
Transmission electron microscopy (TEM) analysis of alloy M4: (**a**–**c**) Bright-field TEM micrographs and corresponding SADPs of borides and matrix: (**d**) corresponding SADPs from primary Cr-rich M_2_B boride reflected from [0$\bar{3}$3], (**e**) corresponding SADPs from eutectic Cr-rich M_2_B boride reflected from [004], (**f**) corresponding SADPs from α-(Fe, Cr) matrix reflected from [002], (**g**) corresponding SADPs from Mo-rich M_2_B boride reflected from [2$\bar{2}$0].

**Figure 5 materials-12-00256-f005:**
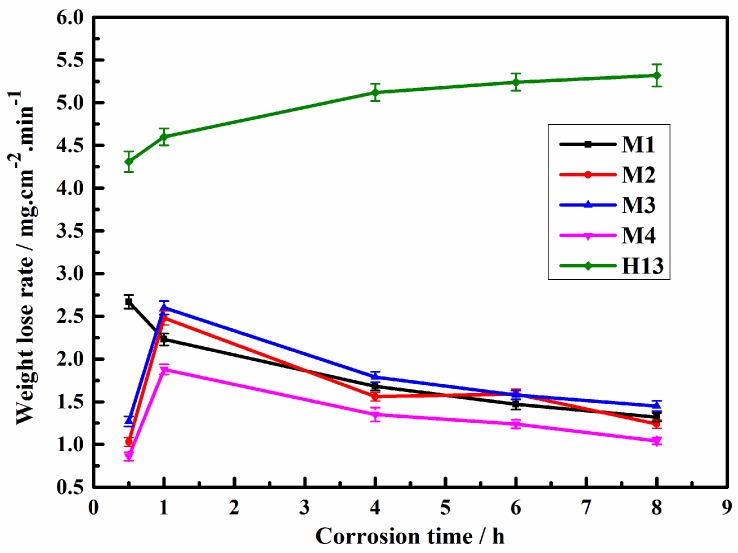
Weight loss per square centimeter of the samples per minute in molten aluminum at 750 °C for different corrosion times.

**Figure 6 materials-12-00256-f006:**
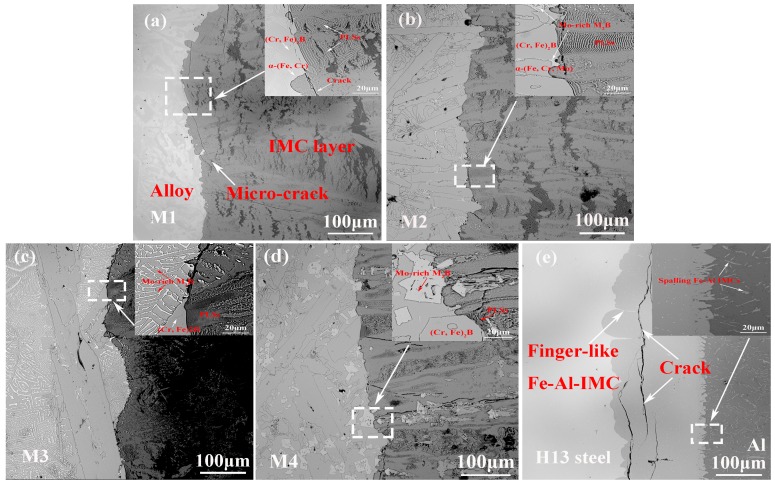
Cross-sectional BSE images of corrosion layers corroded by molten aluminum at 750 °C for 8 h: (**a**) M1; (b) M2; (**c**) M3; (**d**) M4; (**e**) H13. The top right corner images are the high magnification views of the images that are marked with rectangles.

**Figure 7 materials-12-00256-f007:**
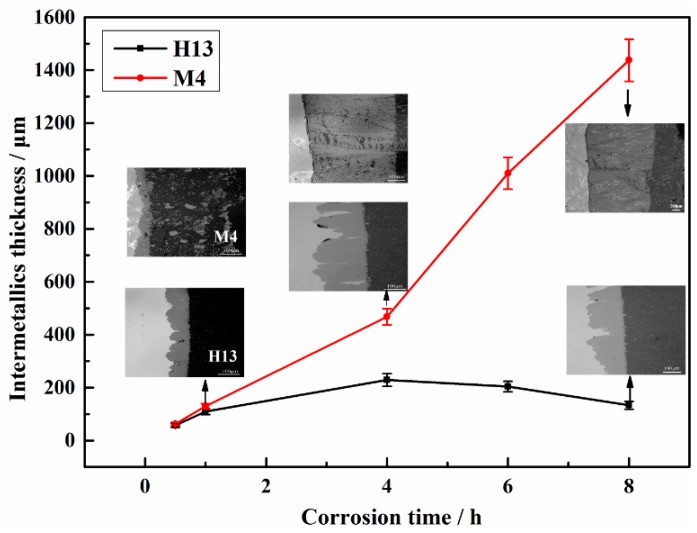
Relationship between the IMC thickness and the corrosion time compared with the interfacial corrosion morphologies of M4 and H13 that were corroded by molten aluminum at 750 °C.

**Figure 8 materials-12-00256-f008:**
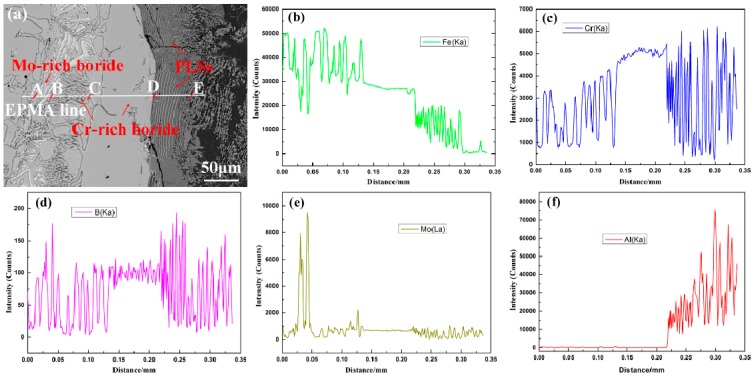
EPMA analysis of the corrosion layer along line segment AE for sample M3 at 750 °C for 8 h in molten aluminum; (**a**) BEI; (**b**) Fe; (**c**) Cr; (**d**) B; (**e**) Mo; (**f**) Al.

**Figure 9 materials-12-00256-f009:**
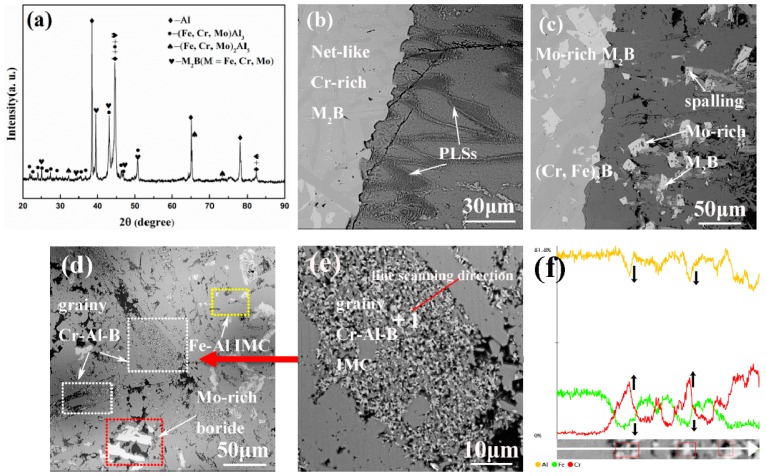
The characterization of the corrosion products layer: (**a**) the XRD pattern of the cross-sectional corrosion layer; (**b**,**c**) BSE images of cross-sectional interfacial layer; (**d**) the longitudinal interfacial layer; (**e**) magnification view of the grainy Cr-Al-B IMCs; (**f**) EDX line analysis in (**e**).

**Figure 10 materials-12-00256-f010:**
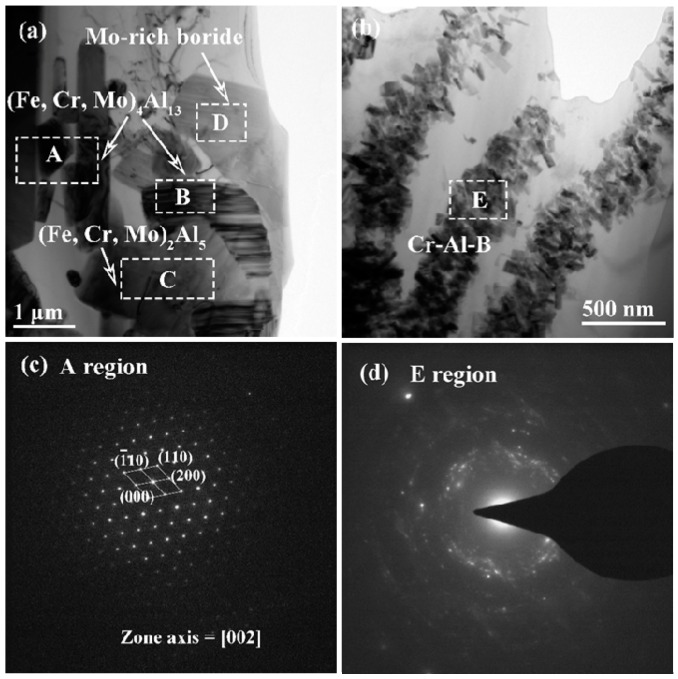
TEM images of the corrosion products of alloy M4 in molten aluminum for 8h: (**a**,**b**) Bright-field TEM micrographs and corresponding SADPs of corrosion products: (**c**) corresponding SADPs from (Fe, Cr, Mo)4Al13 reflected from [002], (**d**) corresponding SADPs from Cr-Al-B IMCs.

**Figure 11 materials-12-00256-f011:**
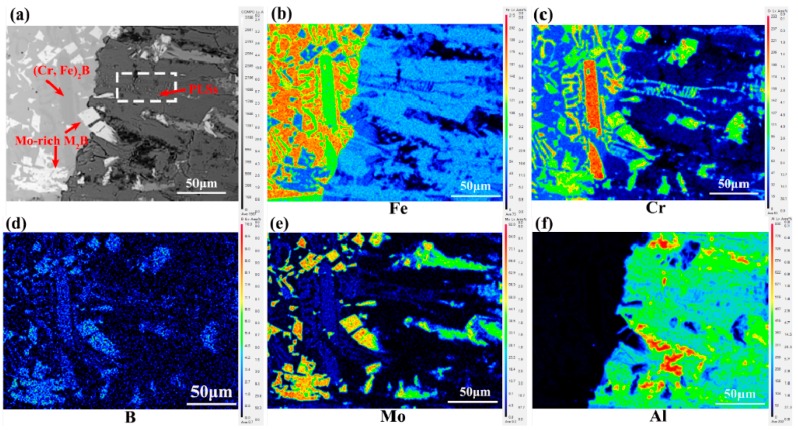
Elemental X-ray mapping distribution of alloy M4 after corrosion test for 8 h at 750 °C: (**a**) interfacial corrosion layer morphology; (**b**) Fe; (**c**) Cr; (**d**) B; (**e**) Mo; (**f**) Al.

**Figure 12 materials-12-00256-f012:**
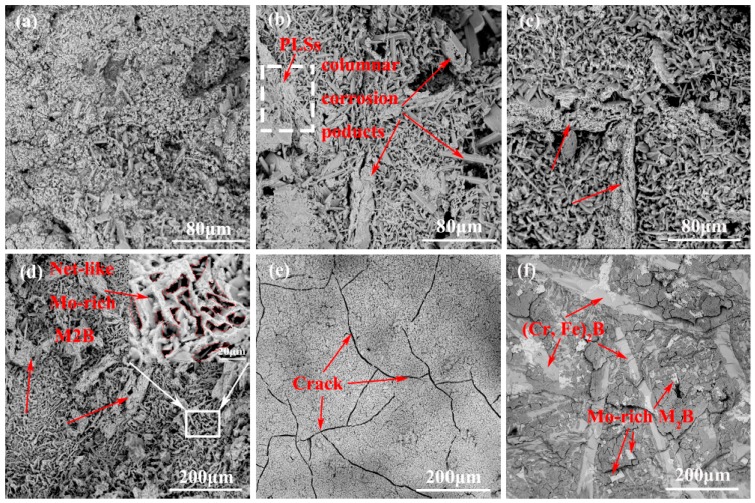
Morphologies of the corroded surface after removing the residual aluminum with a 10% NaOH solution: (**a**) M1; (**b**) M2; (**c**) M3; (**d**) M4; (**e**) H13, and the morphology after removing the IMC layer of M4 (**f**). All the samples were immersed in molten aluminum for 8 h at 750 °C.

**Table 1 materials-12-00256-t001:** Chemical compositions (in wt.%) of Fe-Cr-B cast steels and the H13 steel.

Elements	C	B	Cr	Mo	Si	Al	Mn	Cu	V	Fe
M1	0.25	3.68	13.54	0	0.73	0.44	0.32	0.13	0.07	Bal.
M2	0.26	3.42	14.73	3.22	0.98	0.79	0.31	0.17	0.05	Bal.
M3	0.26	3.12	15.28	6.19	1.13	0.97	0.32	0.25	0.10	Bal.
M4	0.28	3.19	16.35	8.30	1.21	0.85	0.34	0.24	0.07	Bal.
H13	0.38	-	5.30	1.30	1.00	-	0.40	-	0.90	Bal.

**Table 2 materials-12-00256-t002:** The average composition (in at.%) of M_2_B-type boride (M = Fe, Cr, Mo) in Fe-Cr-B cast steels measured by using wavelength dispersive X-ray (WDX) at each point in Figure 3a–d.

Number	Fe	Cr	B	Mo	Calculated Formula
1	37.36 ± 1.41	29.38 ± 0.34	33.26 ± 1.80	0	Fe1.12Cr0.88B
2	44.55 ± 1.24	22.29 ± 1.23	33.16 ± 2.40	0	Fe1.34Cr0.68B
3	27.81 ± 0.95	19.16 ± 0.85	34.74 ± 0.96	18.29 ± 0.78	Fe0.80Cr0.55Mo0.53B
4	25.26 ± 1.05	19.79 ± 0.94	30.94 ± 1.50	24.01 ± 1.30	Fe0.82Cr0.64Mo0.78B
5	21.84 ± 0.80	17.36 ± 0.28	37.11 ± 1.25	23.69 ± 1.45	Fe0.59Cr0.47Mo0.64B

**Table 3 materials-12-00256-t003:** Chemical composition (in at.%) of the phases (in Figure 10a,b) in the corrosion products of M4 analyzed by TEM-EDS.

Regions	Fe	Cr	B	Mo	Al	Phases
A	23.15	1.08	0	0.23	75.54	(Fe, Cr, Mo)_4_Al_13_
B	19.77	1.94	10.26	0.21	67.82	(Fe, Cr, Mo)_4_Al_13_
C	25.19	2.07	0	0.43	72.31	(Fe, Cr, Mo)_2_Al_5_
D	18.30	15.32	43.62	21.64	1.12	Mo-rich boride
E	1.24	22.33	21.08	0	55.32	Cr-Al-B IMC

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
