# Peer review of "The Influence of A Mo Addition on the Interfacial Morphologies and Corrosion Resistances of Novel Fe-Cr-B Alloys Immersed in Molten Aluminum"

_materials, 2019, doi:10.3390/ma12020256_

Reviewer 1 Report

The article is very well thought out and prepared very carefully.

In my opinion, it is suitable for printing.

Author Response

Dear reviewer,

Thank you very much for your valuable review and recognition on our paper. We have revised the manuscript carefully based on the your kind advices.

Reviewer 2 Report

The article describes interesting and important results, but the text itself causes some comments and questions.

- The difference in preparation of samples M1-M4 should be explained in more details;
- XRD data are presented in Fig.2 only, not in Fig. 2 and 3 (line 112);
-  It is not so convenient for readers to get back from table 2 to Fig. 3. The table 2 and it's description should be situated right after Fig. 3;
- There is no accordance between the Fig. 4 and the description of it concerning to Cr and B distribution (Fig. 4c, 4d and lines 128-130);
- If the different colors correspond to different concentrations of elements, in this case it will be more convenient to have the color scale in Fig. 4 and 12;
- In Fig. 6 samples M2, M3, M4 demonstrate the same trend. Why do you separate the M3 sample? (lines 178-179)
- If the M4 sample demonstrates the best corrosion resistance (according to Fig 6), why don't you try to make a sample with a higher content of Mo than M4?
- What is the reason to make some analyses for sample M3 (Fig. 4 and 9) and other analyses for sample M4 (Fig. 5, 8, 10, 11, 12)?
- The sentence in lines 179-181 is not understandable, M1 is mentioned twice.

Author Response

Dear reviewer,

Thank you very much for your valuable review and comments on our paper. We have revised the manuscript carefully based on your kind advices. The details of the modified manuscript are as followed:

- The difference in preparation of samples M1-M4 should be explained in more details;

Reply: Line 71-72: adding “Notably, all the prepared alloys were adopted the multiple pouring in one furnace by adding the charge materials.”
- XRD data are presented in Fig.2 only, not in Fig. 2 and 3 (line 112);

Reply: Line 117: adding “respectively”.
-  It is not so convenient for readers to get back from table 2 to Fig. 3. The table 2 and it's description should be situated right after Fig. 3;

Reply: We have adjusted the position of Fig. 3 and Table 2. the
- There is no accordance between the Fig. 4 and the description of it concerning to Cr and B distribution (Fig. 4c, 4d and lines 128-130);

Reply: Owing to the EPMA analysis of the compositions of various borides in table 2, we have determined to omit the Fig. 4 and corresponding analyses in line 127-131.

Fig.4
- If the different colors correspond to different concentrations of elements, in this case it will be more convenient to have the color scale in Fig. 4 and 12;

Reply: In Fig 12, we have added the color scale.
- In Fig. 6 samples M2, M3, M4 demonstrate the same trend. Why do you separate the M3 sample? (lines 178-179)

Reply: Because of the variation trend in the corrosion time 0.5h and 1h, unlike the Mo containing of M2-M3 alloys, M1 displays a down trending.  
- If the M4 sample demonstrates the best corrosion resistance (according to Fig 6), why don't you try to make a sample with a higher content of Mo than M4?

Reply: Considering the cost of material, we intend to further ameliorate the material properties by improving the morphology and phase distribution, rather than simply adding Mo. Of course, we will consider the influence of higher content of Mo on the corrosion property in the later stage.
- What is the reason to make some analyses for sample M3 (Fig. 4 and 9) and other analyses for sample M4 (Fig. 5, 8, 10, 11, 12)?

Reply: We consider to analysis the sample M3 (Fig. 4) in order to make a direct view the of distribution of B, Mo etc. elements. Owing to the EPMA analysis of the compositions of various borides, we have determined to omit the Fig. 4 and corresponding analyses. As for the Fig. 9, we want to analyze the composition of PLSs. Additionally, the orders of the figures involved have been rearranged.
- The sentence in lines 179-181 is not understandable, M1 is mentioned twice.

Reply: M1-M4 was replaced by “M2 and M3 alloy”.

Reviewer 3 Report

The manuscript entitled: 'Influence of Mo addition on the interfacial morphologies and corrosion resistances of novel Fe-Cr-B alloys immersed in molten aluminum' is written with a specific objective of reducing the corrosion rate of Fe-based alloys in molten Al environment. The addition of B seems to have worked, but  I have the following concerns:

- The XRD pattern of the M4 sample has less intense carbide peaks compared to the other samples, M1, M2, and M3. It should have more carbides since the Mo content is highest in the M4 samples. Such behavior should be carefully explained.

- The changes in the microstructure from M1 to M4 should be explained.

- Since the values in the tables are average values, error bars should be introduced with all numerical data.

- English language should be improved.

Author Response

Dear reviewer, 

Thank you very much for your valuable review and comments on our paper. We have revised the manuscript carefully based on your kind advices. The details of the modified manuscript are as followed :

- The XRD pattern of the M4 sample has less intense carbide peaks compared to the other samples, M1, M2, and M3. It should have more carbides since the Mo content is highest in the M4 samples. Such behavior should be carefully explained.

Reply: Line 124-127, adding the analysis “Additionally, from the XRD pattern of M4 sample, it can be observed that carbide peaks are more intense than that of M1-M3, which may be inferred that there are more carbides formation owing to the highest content among the designed alloys.”

- The changes in the microstructure from M1 to M4 should be explained.

Reply: Line 159-168, we have rewritten the changes of M2B-type borides in M1-M4 alloys.

Reply: “The formation of M2B-type borides is ascribe to ………Obviously, superfluous Mo atoms would may also dissolve in carbide and ferrite.”- Since the values in the tables are average values, error bars should be introduced with all numerical data.

Reply: Error bars have been introduced with all numerical data to the average composition (in at. %) of M2B-type boride (M = Fe, Cr, Mo) in Fe-Cr-B cast steels measured by using WDX at Table 2.

- English language should be improved.

Reply: Details of the modified manuscript:

1.  Page 1, line 13, “were” ® “was”; line 15, “could also”® “also”; line 18, “had a synergistic” ® “which had a synergistic”; line 20, “H13” ® “H13 steel”; line 26, “in the hot dip aluminizing” ® “hot dip aluminizing”; line29, “these components must be changed frequently, which wastes” ® “frequent change of these compoents wastes”; line 36, “between a steel” ® “between steel”; line 39, “the molten aluminum” ® “molten aluminum”, “corrosion resistance” ® “corrosion resistances”; line 40, “resulting” ® “resulted”, “phase transition stress” ® “phase transition stresses”.

2. Page 2, line 44, “surface treatment materials is” ® “surface treatments are”; line 48, “much attention has” ® “much attentions have”; line 51, “has been proven” ® “proved”; line 57-60, “the hardenability co-addition of Mo” ® “the hardenability by co-addition”, “Although the Fe-Cr-B-Mo cast steel with different B additions was studied” ® “Although the Fe-Cr-B-Mo cast steels with different B additions were studied”.

3. Page 3, line 81, “electric balance” ® “electronic balance”; line 83, “being inset into” ® “being inserted into”; line 84, “the uncovered surface was” ® “the uncovered surfaces were”; line 86, “each type of alloy” ® “each type alloy”; line 91, “mg / cm2·min” ® “mg·cm-2·min-1”; line 101, “Additionally” ® “Besides”.

4. Page 4, line 114-115, “3wt.% Mo”® “3.22wt.%Mo”, “6wt% Mo” ® “6.19wt.% Mo”, “9wt.% Mo” ® “8.30wt.% Mo”; line 120-121, “reaches 6.19wt.% Mo”, “reaches 8.3” ® “reaches to 6.19wt.% Mo”, “reaches to 8.30wt.% Mo”.

5. Page 5, line 148, “Notably, when the Cr content” ® “Notably, it has been reported that when the Cr content”.

6. Page 6, line 157, “d(110) = 0.3900” ® “d(110) = 0.3900nm”; line 158, “M2B-type” ® “M2B-type borides”; line 175, “(mg·cm-2·min-1)” was deleted; line177-178, “H13” ® “H13 steel”, “M1-M4 sample” ® “M1-M4 samples”, line 183, “at the immersion time of 8h” ® “at 8h”, line187-188, “H13” ® “H13 steel”.

7. Page 7, line 202, “(Fig. 6e)” was added; line 215, “especially for a Mo-rich M2B-type boride” ® “especially for the Mo-rich M2B-type boride”;

8. Page 8, line 223, “IMC thickness and the corrosion time” ® “the thicknesses of IMCs and the corrosion times”; line 227, “at 750°C” was deleted; line 230, “On the one hand” ® “On one hand”, line 232, “of the substrate” ® “matrix”; line 234, “for the decrease in the IMC thickness of H13” ® “to explain the phenomenon”;

9.  Page 9, line 257, “M4” ® “M4 alloy”; line 260, “M2B boride” ® “M2B-type boride”; line 272, “that there some” ® “that there are some”.

10.  Page 11, line 304, “reveals that both have the effect” ® “reveals that both of them have the effect”.

11. Page 12, line 343, “There are some columnar structures (where the arrows are pointed in Fig. 12) that appeared” ® “Some columnar structures (where the arrows are pointed in Fig. 12) appear”; line 345, “the PLSs are shown in Fig. 12b” ® “the PLSs are also shown in Fig. 12b”; line 354, “the Fe-Al IMCs spalling off” ® “the Fe-Al spallation of Fe-Al IMCs”.

12. Page 13, line 371, “and a blocky hypereutectic” ® “and appear a kind blocky hypereutectic”; line 372, “with the increase in Mo addition” ® “with the increase of Mo addition”; line 377, “a much slower corrosion rate of the matrix” ® “a much slower corrosion rate than the matrix”.
